# Improvement of Gait in Patients with Stroke Using Rhythmic Sensory Stimulation: A Case-Control Study

**DOI:** 10.3390/jcm11020425

**Published:** 2022-01-14

**Authors:** Yungon Lee, Sunghoon Shin

**Affiliations:** 1Research Institute of Human Ecology, Yeungnam University, Gyeongsan-si 38541, Korea; lyg2311@ynu.ac.kr; 2Neuromuscular Control Laboratory, Yeungnam University, Gyeongsan-si 38541, Korea; 3School of Kinesiology, College of Human Ecology & Kinesiology, Yeungnam University, 221ho, 280 Daehak-ro, Gyeongsan-si 38541, Korea

**Keywords:** stroke, gait, rhythmic sensory stimulation, stimulus benefit

## Abstract

Patients with stroke suffer from impaired locomotion, exhibiting unstable walking with increased gait variability. Effects of rhythmic sensory stimulation on unstable gait of patients with chronic stroke are unclear. This study aims to determine the effects of rhythmic sensory stimulation on the gait of patients with chronic stroke. Twenty older adults with stroke and twenty age- and gender-matched healthy controls walked 60 m under four conditions: normal walking with no stimulation, walking with rhythmic auditory stimulation (RAS) through an earphone in the ear, walking with rhythmic somatosensory stimulation (RSS) through a haptic device on the wrist of each participant, and walking with rhythmic combined stimulation (RCS: RAS + RSS). Gait performance in the stroke group significantly improved during walking with RAS, RSS, and RCS compared to that during normal walking (*p* < 0.008). Gait variability significantly decreased under the RAS, RSS, and RCS conditions compared to that during normal walking (*p* < 0.008). Rhythmic sensory stimulation is effective in improving the gait of patients with chronic stroke, regardless of the type of rhythmic stimuli, compared to healthy controls. The effect was greater in patients with reduced mobility, assessed by the Rivermead Mobility Index (RMI).

## 1. Introduction

Stroke is a neurological disorder caused by ischemic and hemorrhagic damage to the cerebrovascular vessels [1,2,3,4,5]. The most common symptom of stroke is a movement disorder defined as loss or limitation of muscle control and motor function [6,7,8,9]. Patients with stroke suffer from impaired locomotion in their daily lives due to movement disorders [10] and exhibit unstable walking with increased gait variability [8,11,12]. Unstable gait with increased gait variability increases the risk of falling [13,14,15,16,17,18]. Falls in patients with stroke is one of the most serious problems, causing reduced mobility, limited function, serious injury, and death [19].

Compared with healthy controls, patients with stroke have decreased gait performance (generally expressed as the mean of spatiotemporal parameters) [20,21,22,23], and increased gait variability (expressed using the standard deviations (SD) or coefficients of variation (CV) of the gait spatiotemporal variables) [13]. Gait performance in individuals with post-stroke hemiparesis is characterized by reduction in the following: walking speed, cadence, stride length, hip joint angle at peak extension, knee joint angle at toe-off or during swinging [24,25,26], and increased foot lateral displacement during swinging [25,27,28]. Changes in walking variability, known as physiological signals that reflect alterations in walking characteristics due to aging and disease [29,30] are pronounced in patients with stroke [13]. For example, Balasubramanian et al. [13] reported that the SD of the spatiotemporal gait parameters, including the step length, swing, and stride time for patients with stroke were greater than those of healthy individuals in the same age group. Kao et al. [14] also reported significant increases in the SD of step length, step width, and margin of stability compared to those of healthy controls, while patients with stroke walked at four different speeds (60%, 80%, 100% of their preferred speeds, and as fast as possible). These results indicate that increased walking variability is closely related to walking in patients with stroke. Therefore, reducing such variability should be considered to improve walking in patients with stroke. 

Rhythmic sensory stimulation, which can be used to improve the gait of persons with stroke, Parkinson’s disease, or the elderly, includes sensory feedback utilizing auditory, somatosensory, and external visual stimuli that provide spatial and temporal information to promote locomotion [31,32,33,34,35,36,37]. Rhythmic auditory stimulation (RAS), which stimulates hearing with fixed rhythms; rhythmic somatosensory stimulation (RSS), which provokes the somatosensory system with rhythmic vibrations; and rhythmic visual stimuli, which stimulate vision with constant patterns, affect the motor system of the human body [33,34,38,39,40,41]. This rhythmic sensory stimulation enhances the walking ability of patients with motion impairments and reduces walking variability. For example, RAS has been shown to increase gait speed [35,42] and stride length [36,43] but reduces the CV of stance time and double support time in a patient with stroke [11]. RSS has been shown to increase the stride length of patients with Parkinson’s disease [44]. Rhythmic auditory or somatosensory input appears to induce a constant locomotion pattern in this gait improvement mechanism by regularly stimulating the central nervous system (CNS), which controls the central pattern generator [45].

Considering these results, applying rhythmic sensory stimulation to patients with stroke could potentially reduce gait variability and improve gait performance [11,46,47,48,49,50,51,52,53]. Nevertheless, the effects of RAS, RSS, or rhythmic combined stimulation (RCS: RAS + RSS) on the gait of patients with stroke are unclear and remain unknown. Similar to previous studies [34,54], which used a mixture of auditory and visual stimulation, the main concern of this study was how the combined effects of auditory and somatosensory stimuli affected the walking of patients with stroke. Therefore, this study aimed to determine whether RAS, RSS, or RCS could improve the walking ability of persons with stroke. We hypothesized that individuals with stroke will have worse walking performance and greater gait variability than healthy controls and that RCS will induce greater gait improvement in individuals with stroke than RAS or RSS alone.

## 2. Materials and Methods

This study was approved by the Bioethics Committee (IRB-2019-04-003-001), and all participants provided written informed consent.

### 2.1. Participants

A total of 40 people were enrolled in the study, including 20 participants with stroke and 20 healthy controls who were matched for gender and age (Figure 1). The Gyeongsan Public Health Center was used to enroll the subjects who had suffered from a stroke. The control group consisted of healthy community-dwelling adults who had similar age, height, weight, and gender to the study participants. The participants in the control groups were recruited through community announcements. Each participant was recruited and registered between April 2019 and November 2019. The experimental period was between July and November. Participants were selected using minimal randomization, and there were no statistically significant variations in age, height, weight, gender, or BMI between the two groups in any of the measured variables (Table 1). Among the patients with stroke, those who were able to move independently and participate willingly in everyday activities were included in the study. In the experiment, patients with stroke who were unable to converse or walk more than 10 m were excluded. In the control group, participants with musculoskeletal diseases or neurological disorders were also excluded from the study. Interviews with patients who had suffered a stroke and their medical records were used to gather specific information about them. Participants gave their informed consent by signing the Institutional Review Board (IRB) consent form and agreeing to take part in the investigation. There were no volunteers who complained of side effects during the experiment, and no adverse events were recorded after the study was completed.

### 2.2. Study Procedures

The Mini-Mental State Exam [55] and Rivermead Mobility Index (RMI) were used to assess the cognitive and motor impairment levels of the participants, respectively [56]. In addition, Timed Up and Go tests were conducted to assess the quickness and dynamic equilibrium of the participants [57]. Thereafter, the participants rested for 5 min and then participated in gait experiments using rhythmic sensory stimulation.

The participants received sufficient explanation from the investigators before participating and performed preliminary exercises for 10 min to familiarize themselves with the experimental protocol. Two 7-axis inertial measurement unit (IMU) sensors were attached to both feet in all patients, who were asked to return from a 30 m “return point” for a total of 60 m of walking. The participants were tested under four conditions (Figure 2 and Figure 3): (1) no stimulation, (2) RAS, (3) RSS, and (4) RCS. Under the first condition, the participants walked normally. The investigator then matched the metronome beat with the cadence collected from the normal walking of the participant and asked the participant to step in time with the metronome beat [44]. Under the second condition, each participant wore earphones and walked to the RAS in time with the metronome beat. Under the third condition, each participant wore a wearable haptic device on the wrist without paralysis and walked to the return point following the vibration of the metronome. Under the fourth condition, each participant wore earphones and the wearable haptic device and walked to the return point in time following the metronome beat and vibration. The last three conditions according to these rhythmic sensory stimuli were performed in random order. The experiment took approximately 50 min, and a 5 min rest period was provided for each experimental condition. Depending on the condition of the participant, the rest time was adjusted to 5–15 min if needed. 

### 2.3. Extraction of Gait Parameters and Providing Rhythmic Sensory Stimulation

Two 7D (3D accelerometer, 3D gyroscope, and 1D barometer) inertial measurement units (IMUs; Physilog 5, Gait Up, Lausanne, Switzerland) that were validated by previous studies [58,59,60,61], particularly for assessing the mobility of patients with stroke [62], were used to obtain the gait measurements. Data were extracted from two 7-axis IMU sensors attached to the feet of the participants and transferred to a spreadsheet file in conjunction with an analysis software. The initial three footprints of the collected data were manually excluded to minimize experimental variations. The mean walking performance of the calculated data was determined, and the variability was determined as the CV value, which was calculated as ((standard deviation/mean) × 100) [30,63].

Rhythmic sensory stimulation was presented as a metronome beat by linking the pulse wearable haptic device (Soundbrenner, Berlin, Germany) and earphones to the tablet PC application software provided by the manufacturer [64]. The wearable haptic device was portable and could be worn on the wrist, whereas the earphone was a small device that could be worn on the ear. The difference between the stimulus condition and the normal condition for each of the gait metrics, i.e., stimulus condition minus normal condition, was used to calculate stimulus benefit in improving walking performance. In addition, the benefit of the stimulus in reducing gait variability was normal minus stimulus condition [65].

### 2.4. Statistical Analysis

The Shapiro–Wilks normality test was performed to confirm that the data in this study were normally distributed. Gait analysis based on rhythmic sensory stimulation was performed by a two-way repeated measures analysis of variance with one between factor and one within factor, and the mean difference was verified at the 0.05 significance level. The independent variables were the group (stroke or healthy control) and type of rhythmic sensory stimulation (no stimulation, RAS, RSS, or RCS), and the dependent variables were the gait performance and variability. When there were interaction effects, three tasks (RAS, RSS, and RCS) were post-tested using the multiple comparisons, with Bonferroni correction. The significance level was corrected to 0.05/6 = 0.0083 by the Bonferroni correction. In addition, the associations between RAS, RSS, and RCS stimulus benefits in gait performance and variability and RMI were analyzed by simple linear regressions, respectively. All data analyses and statistical processes were performed using SPSS ver. 23 (IBM, Armonk, NY, USA) software. The effect size of the two-way repeated measures analysis of variance was calculated as partial eta-squared ($\eta_{p}^{2})$. Here, $\eta_{p}^{2}$ values of 0.01, 0.06, and 0.14 represented small, medium, and large effect sizes, respectively. In the post hoc test for interactions, the effect size was determined as Cohen’s d, where values of 0.2, 0.5, and 0.8 corresponded to small, medium, and large effect sizes, respectively. The sample size of the two-way repeated measures in ANOVA was calculated using G*power 3.1.9.4 software (Heinrich-Heine Düsseldorf University, Düsseldorf, Germany). As a result of setting the effect size (*f*) to 0.25, significance level (α) to 0.05, and power (1–β) to 95% in the number of samples, the optimal sample size was determined to be 38 subjects.

## 3. Results

Table 2 presents the changes in gait performance and variability of the patients with stroke and healthy controls owing to rhythmic sensory stimulation. The gait speed, stride length, cadence, and swing ratio of the gait performance of the group of patients with stroke significantly decreased compared to those of the healthy control group (*p* < 0.05). In addition, the gait cycle, stance ratio, and double support ratio of the gait performance of the group of patients with stroke significantly increased compared to those of the healthy control group (*p* < 0.05). There are significant differences in gait speed, stride length, gait cycle, cadence, and double support ratio according to the rhythmic sensory stimulation (*p* < 0.05), and there are significant differences in gait speed, stride length, and double support ratio according to the interaction (*p* < 0.05).

According to the post hoc test, the stroke group had significantly faster gait speed under the RAS (*p* = 0.001, d = 0.29), RSS (*p* = 0.001, d = 0.30), and RCS (*p* = 0.001, d = 0.31) conditions compared to that for normal walking (Figure 4a), as well as significantly increased stride length under the RAS (*p* = 0.001, d = 0.30), RSS (*p* = 0.001, d = 0.27), and RCS (*p* = 0.001, d = 0.27) conditions compared to that for normal walking (Figure 4b). Moreover, the stroke group showed significantly reduced double support ratios under the RAS (*p* = 0.001, d = 0.28) and RCS (*p* = 0.001, d = 0.22) conditions compared to that for normal walking (Figure 4c).

The CV of stride length, gait cycle, stance ratio, and swing ratio for the patients with stroke were significantly higher than those of the healthy controls (*p* < 0.05). There were significant differences in the CV of stride length, gait cycle, stance ratio, and swing ratio according to the rhythmic sensory stimulation (*p* < 0.05), as well as significant differences in the CV of stride length, gait cycle, stance ratio, swing ratio, and double support ratio according to the interaction (*p* < 0.05).

According to the post hoc test, the stroke group had significantly reduced stride length CV under the RAS (*p* = 0.001, d = 0.77), RSS (*p* = 0.001, d = 0.69), and RCS (*p* = 0.001, d = 0.73) conditions compared to that for normal walking (Figure 5a), as well as significantly reduced gait cycle CV under the RAS (*p* = 0.001, d = 0.75), RSS (*p* = 0.001, d = 0.63), and RCS (*p* = 0.001, d = 0.62) conditions compared to that for normal walking (Figure 5b). In addition, the stroke group had significantly reduced stance ratio CV under the RAS (*p* = 0.001, d = 0.78), RSS (*p* = 0.001, d = 0.72), and RCS (*p* = 0.001, d = 0.79) conditions compared to that for normal walking (Figure 5c), as well as significantly reduced swing ratio CV for the RAS (*p* = 0.001, d = 0.67), RSS (*p* = 0.001, d = 0.55), and RCS (*p* = 0.002, d = 0.65) conditions compared to that for normal walking (Figure 5d). Moreover, the stroke group had a significantly reduced double support ratio CV under the RCS (*p* = 0.002, d = 0.66) condition compared to that for normal walking (Figure 5e). 

However, the gait performance and variability in the healthy control group did not differ significantly under the RAS, RSS, and RCS conditions from those for normal walking (*p* > 0.008). In addition, there were no significant differences between the RAS, RSS, and RCS conditions in the gait variability of the stroke and healthy control groups (*p* > 0.008).

The associations between the RAS, RSS, and RCS stimulus benefits index and RMI in three variables were confirmed because there were significant differences in the averages of speed, stride length, and double support ratio in the group of patients with stroke (Figure 4). There was no association between RAS, RSS, and RCS stimulus benefits in speed, stride length, double support ratio, and RMI.

Lower RMI resulted in significantly greater RAS, RSS, and RCS stimulus benefits in stride length CV (R^2^ = 0.479; *p* = 0.01, R^2^ = 0.420; *p* = 0.01, R^2^ = 0.448; *p* = 0.01). When the RMI was smaller, the RAS stimulation benefits in gait cycle CV were significantly higher (R^2^ = 0.208; *p* = 0.05). As the RMI was lower, the RAS stimulation benefits in stance CV were significantly higher (R^2^ = 0.272; *p* = 0.05). RAS stimulation benefits were significantly stronger in swing CV when RMI was smaller (R^2^ = 0.359; *p* = 0.01) (Figure 6).

## 4. Discussion

This study aimed to investigate the effects of rhythmic sensory stimulation on the gait performance of patients with stroke. Gait data were collected through 7-axis accelerometers under four experimental conditions for patients with stroke and healthy controls. The patients with stroke showed decreased gait performance and increased gait variability while walking compared to the healthy controls. In addition, the gait performance of the group of patients with stroke under the RAS, RSS, and RCS conditions was improved, and the gait variability decreased. The walking performance of the healthy control group did not change under the same conditions. The main results of this study are as follows.

First, RAS improved the gait performance and decreased the gait variability of patients with stroke. Thus, the mean gait speed (8.6%) and stride length (7.6%), which are gait performance parameters, increased with RAS during walking compared to those of normal walking without stimulation. The double support ratio (6.6%) decreased. In addition, when RAS was applied during walking, the gait variability parameters, i.e., the CV of stride length (25.3%), gait cycle (28.1%), stance ratio (21.7%), and swing ratio (25.9%), decreased. Similarly, previous studies reported that RAS improves gait performance and reduces variability in patients with stroke [8,11,35,36,66]. For example, Thaut et al. [8] found that RAS increases the mean gait speed and stride length in relation to the walking performance of patients with stroke. Wright et al. [11] found that RAS reduces the CV of stance time and double support time in gait variability in a patient with stroke. This improvement in walking ability results from regular modulation of the central pattern generator of RAS [67,68]. The regular pattern of rhythmic locomotion of the human body is controlled by the CNS [45]. During walking, RAS is thought to help stabilize gait by inducing a certain movement pattern by periodically stimulating the CNS [67]. In particular, the three rhythmic sensory stimuli in this study showed significant effects in the five gait variability parameters and three gait performance parameters. These results demonstrate that rhythmic sensory stimulation is more effective at reducing gait variability than at improving overall gait performance in patients with stroke.

Second, RSS improved the gait performance and decreased the gait variability of the patients with stroke. The mean gait speed (8.6%) and stride length (7.6%), which are gait performance parameters, increased with RSS during walking compared to those of normal walking without stimulation in patients with stroke. In addition, when RSS was provided during walking, the gait variability parameters, i.e., the CV of stride length (23.7%), gait cycle (24.5%), stance ratio (22.0%), and swing ratio (24.5%), decreased. Similarly, previous studies of patients with Parkinson’s disease and older adults who experienced falls reported that RSS improved gait performance and reduced gait variability in both groups [33]. For example, van Wegen et al. [44] found that RSS increases the mean step length in walking performance in patients with Parkinson’s disease. Galica et al. [33] argued that RSS reduces the SD of gait variability, gait cycle, stance time, and swing time of older persons with fall histories. Previous studies also indicated that a reduction in proprioception increases gait variability and leads to unstable gait [69,70]. RSS has been shown to induce a constant motor pattern by periodically promoting somatosensory input in patients with stroke [67]. Similarly, in this study, RSS helped induce a stable gait in patients with stroke. This is the first study to verify the positive effect of RSS on the gait of patients with stroke.

Third, in this study, RCS improved the gait performance and decreased the gait variability of patients with stroke. The gait performance parameters, mean gait speed (8.6%) and stride length (7.6%) increased, and the double support ratio (5.3%) decreased with RCS compared to those of the natural gait without stimulation. However, when RCS was provided during walking, the gait variability parameters, i.e., stride length (25.1%), gait cycle (24.5%), stance ratio (22.7%), swing ratio (26.0%), and double support ratio (17.3%) decreased. These results were somewhat different from those of previous studies on the effects of mixed stimulation in patients with Parkinson’s disease. For example, Arias and Cudeiro [34] found that mixed stimulation of auditory and visual information did not improve the mean speed in the walking performance of patients with Parkinson’s disease. Suteerawattananon et al. [54] reported that mixed stimulation of auditory and visual information did not increase the mean stride length related to gait performance in patients with Parkinson’s disease. It is not possible to distinguish clearly whether these conflicting results are due to the different subjects using the rhythmic sensory stimuli or the different combinations of sensory stimuli, because the subjects of the previous studies by Arias and Cudeiro [34] and Suteerawattananon et al. [54] were patients with Parkinson’s disease, and the stimuli were different. Therefore, follow-up studies on the effects of various forms of rhythmic sensory stimulation on gait are required.

Fourth, association between RMI and stimulus benefits in gait variability is obvious and RAS is the most effective way but the combination effect between stimulations is not shown. RMI is known as a useful classification criterion to effectively evaluate functional ambulation of stroke patients [71,72]. Gait variability is a quantifiable feature of walking that is altered (both in terms of magnitude and dynamics) in clinically relevant syndromes, such as falling, frailty, and neurodegenerative disease (e.g., Parkinson’s, stroke, and Alzheimer’s disease) [13,16,18,63]. Previous studies have suggested that measures of gait variability may be more closely related to falls, a serious consequence of many gait disorders, than measures based on mean values of other gait parameters such as gait speed, stride length, and stride time [16,73]. Therefore, our study results support previous studies that RMI should be used more extensively to evaluate the physical function related with mobility according to the disability grade of stroke patients, and the effect of therapeutic treatment to improve mobility in stroke patients could be evaluated through the gait variability [71,74].

Finally, our results suggest that rhythmic sensory stimulation treatment can be effective for fall prevention regardless of whether or not long-term rehabilitation training is useful for persons with severely limited function improvement. The fact that the rhythmic sensory stimulation effect during walking is more pronounced in the more severe disorder underscores the usefulness of rhythmic sensory stimulation as a fall prevention tool, as well as a therapeutic tool for patients with severe motor impairment. It is well known that the process of sensory reweighting of persons with stroke during postural control is described such that sensory information that is more reliable or available is weighted more strongly by the brain than less reliable or absent sensory information [75]. However, our study shows that sensory stimulation information during gait as compensation for damaged brain function from stroke is always valid regardless of the type of information and can be used in various ways to prevent falls. In particular, the fact that combined use of RAS and RSS do not have an extra effect and the effect does not accumulate linearly with sufficient sensitivity, a detailed follow-up study on how to maximize its effect is essential. Furthermore, because cognitive recovery in patients with stroke is so important in daily activities [76], the effect of rhythmic sensory stimulation in experimental protocols such as dual-tasking should be thoroughly validated.

### Study Limitations

First, the severity of patients with stroke was evaluated as RMI in this study; however, only one stroke group was considered. Stroke severity could affect the sensitivity of the sensory receptors that utilize the sensory signals in the gait process. Second, the walking speed of the patients with stroke was not sufficiently considered. Understanding how rhythmic sensory stimulation affects patient gait at various gait speeds can provide adequate information for designing an optimal program that will further improve the gait abilities of patients with stroke during rehabilitation. For reference, this study matched the cadence and metronome beats of the rhythmic sensory stimulation in the natural gait of the patients with stroke; however, the gait change of these patients may differ at different gait speeds. Finally, because we derived the study results from a small sample size, generalizing the study results may be difficult. As a result, further validation of the effect of rhythmic sensory stimulation in patients with stroke is required in future studies.

## 5. Conclusions

Treatment methods using rhythmic sensory stimulation improved gait performance and decreased gait variability of patients with stroke, regardless of the type of rhythmic stimuli. These effects were observed despite the increases in walking speed or stride length of the patients with stroke. Rhythmic sensory stimulation was more effective at stabilizing the walking of these patients compared to that of the healthy control group. In particular, this study showed that these benefits are greater in patients with stroke with a relatively high risk of falling.

## Figures and Tables

**Figure 1 jcm-11-00425-f001:**
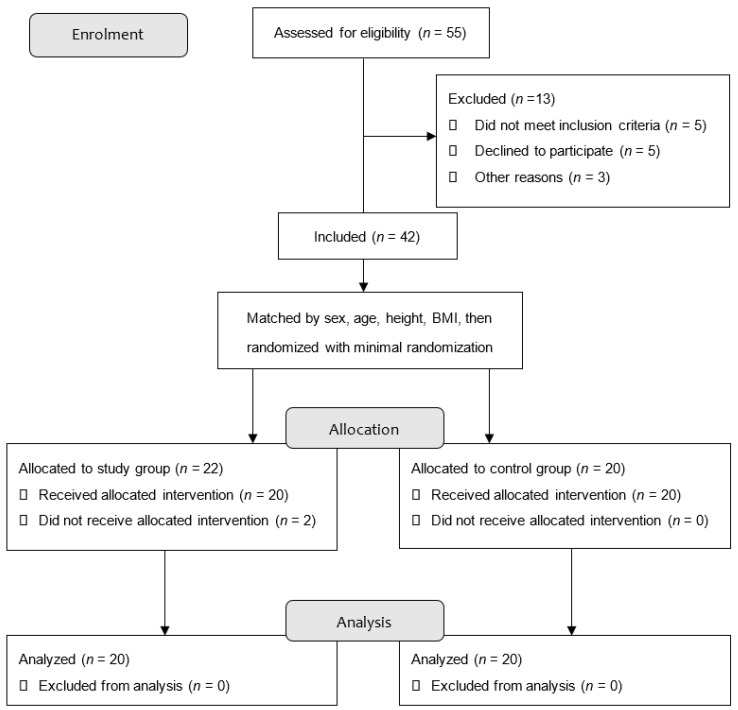
Flow diagram showing experimental design.

**Figure 2 jcm-11-00425-f002:**
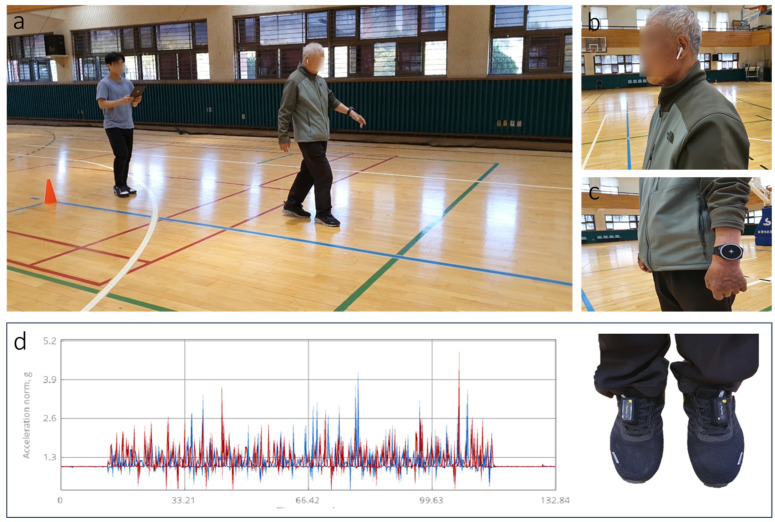
Example of a gait protocol using rhythmic sensory stimulation. (**a**) Experimental scene of a patient with stroke during a 60 m walk; (**b**) rhythmic auditory stimulation (RAS) condition; (**c**) rhythmic somatosensory stimulation (RSS) condition; (**d**) raw data extracted from 7D IMU sensor.

**Figure 3 jcm-11-00425-f003:**
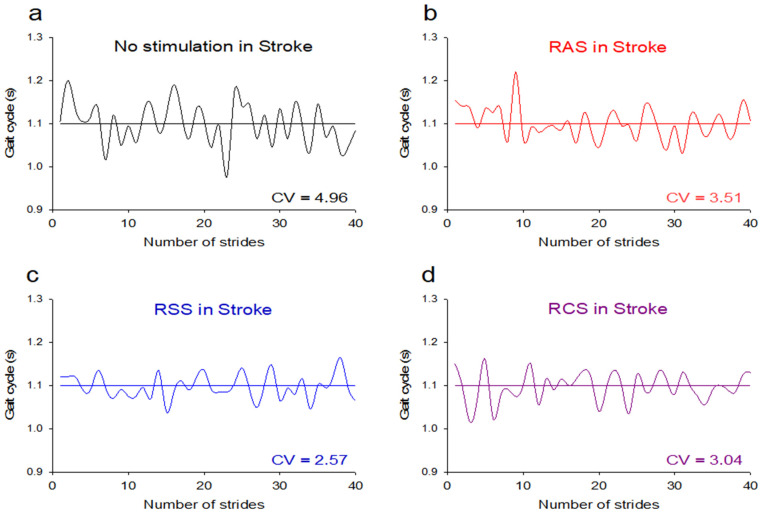
Examples of gait cycle from a patient with stroke in all walking conditions. (**a**) No stimulation: normal walking; (**b**) RAS: rhythmic auditory stimulation; (**c**) RSS: rhythmic somatosensory stimulation; (**d**) RCS: rhythmic combined stimulation (RAS + RSS). The horizontal lines are the average gait cycles of patients with stroke. CV: coefficient of variation.

**Figure 4 jcm-11-00425-f004:**
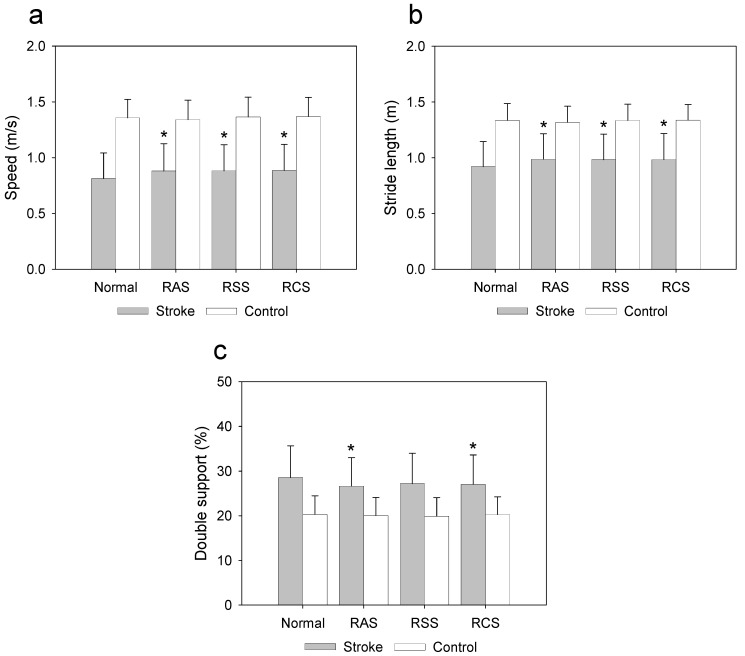
Effects of rhythmic sensory stimulation on gait performance compared to normal walking: (**a**) speed; (**b**) stride length; (**c**) double support. Normal: no stimulation; RAS: rhythmic auditory stimulation; RSS: rhythmic somatosensory stimulation; RCS: rhythmic combined stimulation (RAS + RSS); * indicates significant difference compared to normal condition (*p* < 0.008).

**Figure 5 jcm-11-00425-f005:**
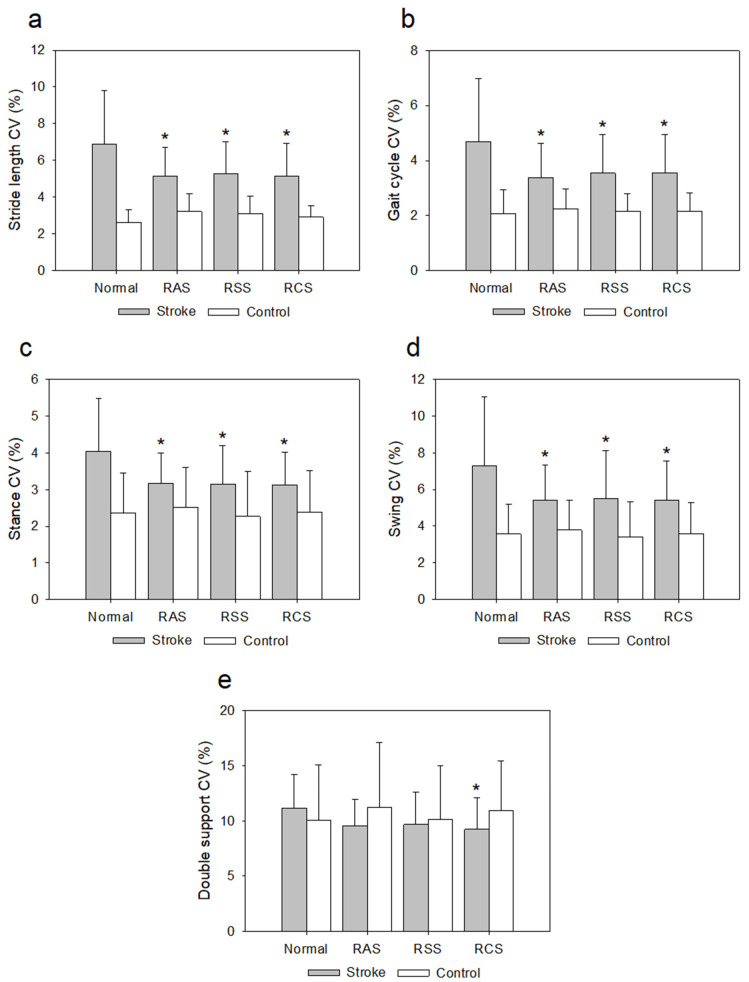
Effects of rhythmic sensory stimulation on gait variability compared to normal walking. Coefficients of variation (CV) of (**a**) stride length; (**b**) gait cycle; (**c**) stance; (**d**) swing; and (**e**) double support. Normal: no stimulation; RAS: rhythmic auditory stimulation; RSS: rhythmic somatosensory stimulation; RCS: rhythmic combined stimulation (RAS + RSS); * indicates significant difference compared to normal condition (*p* < 0.008).

**Figure 6 jcm-11-00425-f006:**
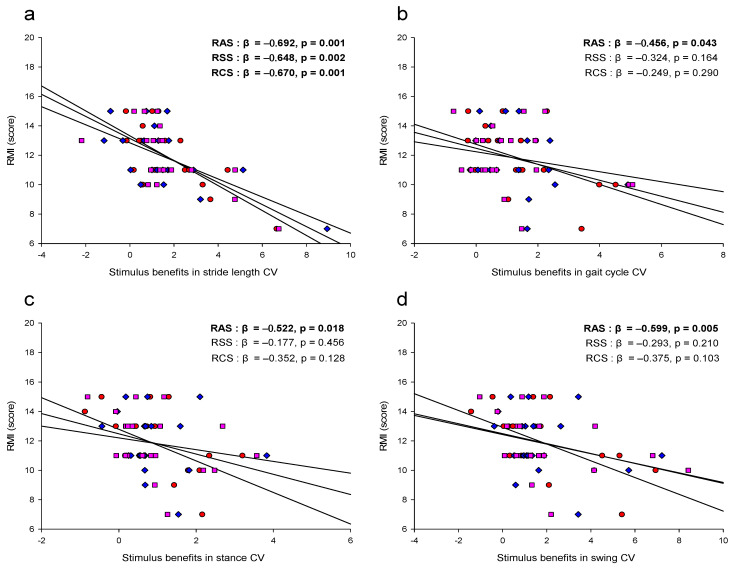
Association between RMI and stimulus benefits in gait variability. (**a**) stimulus benefits in stride length CV; (**b**) stimulus benefits in gait cycle CV; (**c**) stimulus benefits in stance CV; (**d**) stimulus benefits in swing CV. Bold marks indicate significant differences (*p* < 0.05). The stimulus benefit in gait variability is normal condition minus stimulus condition. The red scatter plot is the stimulus benefit for RAS, blue is the stimulus benefit for RSS, and purple is the stimulus benefit for RCS.

**Table 1 jcm-11-00425-t001:** Characteristics of patients in stroke and control groups.

	Stroke (*n* = 20)	Control (*n* = 20)	*p*-Value
Age (years)	72.10 ± 7.15	72.65 ± 6.93	NS
Height (cm)	162.30 ± 8.65	163.60 ± 8.09	NS
Weight (kg)	63.55 ± 8.65	65.51 ± 10.04	NS
Gender (female/male)	8/12	8/12	NS
BMI (kg/m^2^)	23.97 ± 2.52	24.44 ± 2.89	NS
WHR (waist/hip ratio)	0.87 ± 0.05	0.89 ± 0.05	NS
Education (years)	16.50 ± 7.03	16.55 ± 5.24	NS
MMSE ^†^ (Max score: 30)	24.15 ± 4.90	28.45 ± 1.53	<0.001 *
RMI ^‡^ (Max score: 15)	11.85 ± 2.10	15.00 ± 0.00	<0.001 *
Timed Up and Go test (s)	17.32 ± 7.19	7.76 ± 1.13	<0.001 *
No. of falls in previous year (No (%))	0.55 ± 0.99 (35%)	0.05 ± 0.22 (5%)	0.040 *
Period after stroke onset (months)	114.95 ± 78.23		
Hemiparetic side (right/left)	6/14		
Type of stroke (I/H) ^§^	16/4		
Use of walking aid (yes/no)	7/13		

Data are presented as mean ± SD; ^†^ MMSE: Mini-Mental State Exam; ^‡^ RMI: Rivermead Mobility Index; ^§^ I: ischemic; H: hemorrhagic; * indicates significant difference between groups (*p* < 0.05); NS: not significant.

**Table 2 jcm-11-00425-t002:** Results of gait parameters on walking condition in stroke and control groups.

		Walking Condition					
Gait Variable	Group	Normal ^†^	RAS ^‡^	RSS ^§^	RCS ^¶^	*p*-Value	$\eta_{p}^{2}$
Speed (m/s)	Stroke (*n* = 20)	0.81 ± 0.23	0.88 ± 0.24	0.88 ± 0.23	0.88 ± 0.23	Group < 0.001 *	0.607
	Control (*n* = 20)	1.35 ± 0.16	1.34 ± 0.17	1.36 ± 0.17	1.36 ± 0.17	Condition < 0.001 *	0.158
						Interaction 0.002 *	0.143
Stride length (m)	Stroke (*n* = 20)	0.91 ± 0.22	0.98 ± 0.22	0.98 ± 0.23	0.98 ± 0.23	Group < 0.001 *	0.490
	Control (*n* = 20)	1.33 ± 0.14	1.31 ± 0.14	1.33 ± 0.14	1.33 ± 0.14	Condition < 0.001 *	0.163
						Interaction < 0.001 *	0.232
Gait cycle (s)	Stroke (*n* = 20)	1.18 ± 0.20	1.16 ± 0.20	1.16 ± 0.20	1.15 ± 0.19	Group < 0.001 *	0.260
	Control (*n* = 20)	0.99 ± 0.06	0.99 ± 0.07	0.99 ± 0.07	0.98 ± 0.07	Condition 0.023 *	0.096
						Interaction 0.257	0.035
Cadence (steps/min)	Stroke (*n* = 20)	104.10 ± 15.60	105.59 ± 15.37	105.95 ± 15.28	106.32 ± 14.95	Group < 0.001 *	0.305
	Control (*n* = 20)	121.03 ± 8.17	121.33 ± 9.37	121.99 ± 9.68	122.22 ± 9.86	Condition 0.014 *	0.104
						Interaction 0.581	0.015
Stance (%)	Stroke (*n* = 20)	63.19 ± 4.06	62.61 ± 3.52	62.58 ± 4.08	62.49 ± 3.80	Group 0.010 *	0.163
	Control (*n* = 20)	60.11 ± 2.39	60.03 ± 2.16	59.89 ± 2.53	60.06 ± 2.35	Condition 0.126	0.052
						Interaction 0.326	0.029
Swing (%)	Stroke (*n* = 20)	36.80 ± 4.06	37.38 ± 3.52	37.41 ± 4.08	37.50 ± 3.80	Group 0.010 *	0.163
	Control (*n* = 20)	39.88 ± 2.39	39.96 ± 2.16	40.10 ± 2.53	39.93 ± 2.35	Condition 0.126	0.052
						Interaction 0.326	0.029
Double support (%)	Stroke (*n* = 20)	28.54 ± 7.07	26.65 ± 6.36	27.20 ± 6.81	27.01 ± 6.61	Group < 0.001 *	0.314
	Control (*n* = 20)	20.28 ± 4.18	20.03 ± 4.10	19.89 ± 4.19	20.27 ± 3.95	Condition < 0.001 *	0.157
						Interaction 0.006 *	0.106
Stride length CV (%)	Stroke (*n* = 20)	6.87 ± 2.92	5.13 ± 1.59	5.24 ± 1.75	5.14 ± 1.78	Group < 0.001 *	0.488
	Control (*n* = 20)	2.60 ± 0.71	3.20 ± 0.95	3.10 ± 0.96	2.91 ± 0.59	Condition 0.008 *	0.119
						Interaction < 0.001 *	0.286
Gait cycle CV (%)	Stroke (*n* = 20)	4.69 ± 2.28	3.37 ± 1.25	3.54 ± 1.39	3.54 ± 1.41	Group < 0.001 *	0.357
	Control (*n* = 20)	2.05 ± 0.87	2.23 ± 0.71	2.15 ± 0.63	2.14 ± 0.67	Condition < 0.001 *	0.151
						Interaction < 0.001 *	0.218
Stance CV (%)	Stroke (*n* = 20)	4.04 ± 1.42	3.16 ± 0.82	3.15 ± 1.05	3.12 ± 0.88	Group 0.003 *	0.213
	Control (*n* = 20)	2.36 ± 1.09	2.50 ± 1.09	2.26 ± 1.24	2.37 ± 1.13	Condition < 0.001 *	0.129
						Interaction < 0.001 *	0.138
Swing CV (%)	Stroke (*n* = 20)	7.29 ± 3.73	5.40 ± 1.91	5.50 ± 2.63	5.39 ± 2.15	Group < 0.001 *	0.247
	Control (*n* = 20)	3.56 ± 1.64	3.78 ± 1.64	3.41 ± 1.88	3.59 ± 1.71	Condition < 0.001 *	0.162
						Interaction < 0.001 *	0.177
Double support CV (%)	Stroke (*n* = 20)	11.16 ± 3.05	9.54 ± 2.43	9.65 ± 2.91	9.22 ± 2.85	Group 0.575	0.008
	Control (*n* = 20)	10.05 ± 5.03	11.24 ± 5.89	10.10 ± 4.92	10.92 ± 4.53	Condition 0.280	0.033
						Interaction 0.002 *	0.124

Data are mean ± SD; ^†^ Normal: no stimulation; ^‡^ RAS: rhythmic auditory stimulation; ^§^ RSS: rhythmic somatosensory stimulation; ^¶^ RCS: rhythmic combined stimulation (RAS + RSS); * indicates significant difference (*p* < 0.05).

## Data Availability

The datasets used during the current study are available from the corresponding author on reasonable request.

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
