# Peer review of "Improvement of Gait in Patients with Stroke Using Rhythmic Sensory Stimulation: A Case-Control Study"

_jcm, 2022, doi:10.3390/jcm11020425_

Round 1

Reviewer 1 Report

Title

The title must include that this study is observational of cases and controls

Introduction

The introduction should include more information about stroke, such as the most affected population or some risk factors.

The references used in the introduction are too old, it would be advisable to incorporate more up-to-date references.

It would be necessary to include an explanation about the mechanisms by which this change in gait occurs in patients with stroke.

What proportion of stroke patients currently suffer falls

Materials and Methods

In this section I would like the authors to give me more information about the inclusion and exclusion criteria of the participants.

Where was the sample obtained for this study?

Could the authors give more information about the information that the participants received before enrolling in this study?

How long did it take you to recruit that sample?

After a stroke, the gait pattern of a patient tends to become chronic and normalize, which was the time that they considered "optimal" in their study 

The sample size is insufficient to be able to generalize these results 

Reviewer 2 Report

This is an interesting study that investigates whether RAS, RSS, or RCS could improve the walking ability of persons with stroke.

On Page 2, the authors state that "We predicted that individuals with stroke will have..." Suggest changing it to, "We hypothesize that ..."

2. Methods - Please provide more information on how patients were screened, recruited in this study. What was the sampling strategy? Were consecutive patients selected, or randomly recruited?

It is not clear if any patient dropped out of the study due to adverse events?

Data on adverse events, if applicable, should be provided.

It is not clear why regression analyses were not performed to explore the association with outcomes?

Stroke severity plays an imp role in outcomes after stroke - modulating the effect of interventions (https://pubmed.ncbi.nlm.nih.gov/29133696/) Please provide stroke severity data if available.

The effects of sensory stimulation on cognitive outcomes after stroke is another area of interest, and may also be discussed in the Discussion.

.,

Reviewer 3 Report

The article titled: Improvement of Gait in Patients with Stroke using Rhythmic 

Sensory Stimulation is very interesting with practical value. The weak point of this study is very small group of patients. It is very difficult to create any conclusion on such low amount of patients.

Remarks: please expand information about study group, concomitant diseases and treatment, kind of stroke that can also impact results.

What about control group? any diseases, drugs or other medical problems.

Have you any conclusions depending on the side of paresis after stroke?

Round 2

Reviewer 2 Report

No further comments

Reviewer 3 Report

In my point of view manuscript is much improved.